# (7E)-7,8-Dehydroheliobuphthalmin from *Platycladus orientalis* L.: Isolation, Characterization, and Hair Growth Promotion

**DOI:** 10.3390/ijms26115189

**Published:** 2025-05-28

**Authors:** Zikai Lin, Yan Sun, Chengzhao Li, Xiaowei Zhou, Yuting Guo, Zhenhua Wang, Gang Li

**Affiliations:** 1Center for Mitochondria and Healthy Aging, College of Life Sciences, Yantai University, Yantai 264003, China; lzk1215741582@163.com (Z.L.); lichengzhao1997@163.com (C.L.); 2College of Traditional Chinese Medicine, Shandong University of Traditional Chinese Medicine, Jinan 250355, China; sdytsy4023@163.com (Y.S.); zhmqlw@sina.com (Y.G.); 3Xinjiang Production and Construction Corps Key Laboratory of Protection and Utilization of Biological Resources in Tarim Basin, College of Life Science, Tarim University, Alar 843300, China; xiaoweiz1223@163.com

**Keywords:** *Platycladus orientalis* L., androgenetic alopecia, medium-pressure liquid chromatography, (7E)-7,8-Dehydroheliobuphthalmin, Wnt/β-catenin

## Abstract

Androgenetic alopecia (AGA) is a prevalent form of non-scarring hair loss, affecting approximately 32.13% of the population. Seborrheic alopecia is the most frequently observed among its various types, contributing to over 25% of hair loss cases in men. Identifying effective natural compounds or therapeutic agents that stimulate hair growth remains a key research focus. *Platycladus orientalis* L., known for its medicinal properties, shows potential in promoting hair darkening and regeneration, although its mechanisms remain unclear. In this study, Fr2 of *Platycladus orientalis* L. was found to significantly enhance hair growth in mice. Similarly, (7E)-7,8-Dehydroheliobuphthalmin (DHHB) was successfully isolated and purified for the first time through a combination of medium-pressure liquid chromatography and two-dimensional high-performance liquid chromatography. In an alopecia areata (AGA) model using dermal papilla cells (DPCs), DHHB was found to significantly promote cell proliferation and differentiation by down-regulating the expression of androgen receptor (AR) proteins, and activating the Wnt/β-catenin signaling pathway, as compared with the dihydrotestosterone-induced model group. These results indicate that DHHB is a major bioactive compound in *Platycladus orientalis* L. and represents a promising candidate for promoting hair growth.

## 1. Introduction

Androgenetic alopecia (AGA) is a chronic and widespread condition that becomes more prevalent with advancing age. Studies estimate that AGA affects approximately 73% of men and 57% of women over the age of 80, with around 58% of men over 50 also showing signs of the disorder [1]. While AGA does not typically threaten physical health, it is of clinical significance due to its substantial impact on mental well-being, frequently leading to psychological distress and, in some cases, serious mental health disorders among those affected [2,3]. Moreover, AGA has been shown to significantly reduce the quality of life in individuals affected by the condition [4]. The chronic and progressive course of AGA is largely attributed to dihydrotestosterone (DHT), the biologically active metabolite of testosterone, which possesses higher potency and a stronger affinity for androgen receptors than testosterone itself. DHT promotes the miniaturization of hair follicles, progressively diminishing hair growth. This gradual process converts terminal hairs into fine vellus hairs, ultimately culminating in evident hair loss [5,6].

*Platycladus orientalis* L., an evergreen tree belonging to the Cypress family and the Platycladus genus [7], is widely distributed throughout northern and northwestern China. Celebrated as a symbol of vitality and longevity, this culturally and medicinally significant plant has been cultivated for centuries [8]. Its therapeutic properties were first recorded in the ancient herbal classic Shen Nong Ben Cao Jing during the Eastern Han dynasty, where it was credited with cooling the blood, stopping bleeding, clearing heat to relieve coughing, dispelling wind and dampness, reducing toxin-induced swelling, and promoting hair growth and darkening. Drawing from these historical accounts, modern researchers have extensively explored the medicinal potential of *P. orientalis*, revealing promising results and treating various health conditions [9,10,11]. Recent progress in biological and medical research has uncovered that *Platycladus orientalis* L. is rich in multiple bio-active constituents, including flavonoids and cedarol. These compounds have been associated with numerous health-promoting properties, such as antioxidant, neuroprotective, anti-aging, and anti-hyperlipidemic effects [12,13,14].

Numerous studies have documented the hair-growth-promoting properties of *Platycladus orientalis* L. leaves, with a predominant focus on the hot water extract, which is rich in highly polar compounds [15,16,17]. However, recent studies suggest that the volatile oil compounds of *Platycladus orientalis* L. also demonstrate significant effectiveness in promoting hair growth [18]. This indicates the existence of other small polar bioactive molecules in the leaves of *Platycladus orientalis* L. that could enhance hair growth or regeneration. Compared to common drugs available in the market, such as minoxidil and finasteride, the components in natural plants, such as siderophores, have the advantages of fewer side effects and lower therapeutic cost [19]. In this study, medium-high pressure liquid chromatography was used to isolate the chemical constituents from *P. orientalis*. Both an AGA mouse model and DPCs were employed to examine the hair-growth-promoting effects of these compounds and to explore the underlying mechanisms.

## 2. Results

### 2.1. Sample Pretreatment of Platycladus orientalis L.

The powdered branches of *Platycladus orientalis* L. were extracted using n-hexane and the resulting filtrate was concentrated to obtain a crude extract. To concentrate the active compounds, the crude extract was processed using an MCI GEL CHP20P medium-pressure column, resulting in two distinct fractions, labeled Fr1 and Fr2, based on the elution order of their peaks (Figure 1A). These fractions were then isolated by decompression drying, yielding 6.48 g for Fr1 and 5.16 g for Fr2. As shown in Figure 1B, comparing the crude extract and the two fractions revealed minimal overlap between Fr1 and Fr2, facilitating further isolation and purification.

### 2.2. The Isolation and Purification of Fr211

The preparative chromatogram of Fr2 is shown in Figure 2A. When monitored at 254 nm, two fractions, Fr21 and Fr22, were collected. These fractions were co-analyzed with Fr2 using a Kromasil 100-5 Phenyl column to evaluate the effectiveness of the Fr2 preparation, as shown in Figure 2B. The analysis revealed minimal overlap between Fr21 and Fr22, suggesting that Fr22 effectively enriched the primary components of Fr2 (highlighted in a red box). In comparison, lower-content components were concentrated in Fr21 (highlighted in a blue box). In vivo testing demonstrated that Fr22 significantly outperformed Fr21 in promoting hair growth in mice (Figure 3). This finding led to further purification of Fr22 to isolate higher-purity compounds. Using the Kromasil 100-5 Phenyl preparative column, Fr22 underwent further purification (Figure 2C). Upon increasing the sample load, a minor chromatographic peak appeared before the main component of Fr22, though it did not affect the purification process. The collected and concentrated eluted fractions yielded 0.36 g of Fr211, with a purity exceeding 98% (Figure 2D).

Through the purification of Fr211, a single compound was successfully isolated. Based on its physicochemical properties and NMR spectral data (provided in the Appendix A), the structure of this compound was identified as (7E)-7,8-Dehydroheliobuphthalmin. The compound appeared as a pale yellow oil, with an ESI-MS *m*/*z* of 435.10 [M^+^Na]^+^, a chemical formula of C22H20O8, and a molecular weight of 412.12 (Figure 2E). This compound has previously been reported in *Biota orientalis* and *Heliopsis helianthoides var. scabra* [13,20].

^1^H NMR (600 MHz, CD_3_OD): δ 7.55 (^1^H, s, H-7), 6.66 (^1^H, d, *J* = 7.9 Hz, H-6), 6.59 (^1^H, d, *J* = 8.1 Hz, H-3′), 6.51 (^1^H, d, *J* = 1.4 Hz, H-3), 6.42 (^1^H, d, *J* = 1.5 Hz, H-6′), 6.36 (^1^H, dd, *J* = 7.9,1.6 Hz, H-4′), 6.03 (^1^H, dd, *J* = 7.9, 1.4 Hz, H-5), 5.91 (^2^H, d, *J* = 0.9 Hz, H-10), 5.87 (^2^H, d, *J* = 0.9 Hz, H-10′), 4.08 (^1^H, dd, *J* = 10.2, 4.7 Hz, H-8′), 3.73 (^3^H, s, 9-OCH_3_), 3.62 (^3^H, s, 9′-OCH_3_), 3.23 (^1^H, dd, *J* = 13.8, 4.7 Hz, H-7′), 2.77 (^1^H, dd, *J* = 13.8, 10.2 Hz, H-7′). ^13^C NMR (151 MHz, CD_3_OD): δ 172.3 (C-9′), 166.6 (C-9), 147.6 (C-1), 147.3 (C-1′), 147.0 (C-2), 145.5 (C-2′), 141.7 (C-7), 132.4 (C-4′), 128.7 (C-8), 128.2 (C-4), 122.9 (C-5), 122.0 (C-5′), 109.2 (C-3′), 108.3 (C-3, C-6), 107.7 (C-6′), 101.3 (C-10), 100.6 (C-10′), 52.0 (OCH_3_-9′, OCH_3_-9), 44.7 (C-8′), 35.1 (C-7′).

### 2.3. The Effects of Fr1 and Fr2 on AGA Mice

#### 2.3.1. Effects of Fr1 and Fr2 on Hair Growth in Mice

The rate and extent of dorsal hair growth in mice are the most direct and observable measures of a compound’s effectiveness in promoting hair growth. Throughout this study, no allergic reactions or skin damage were noted on the backs of the treated mice. On Day 0, the skin of all mice appeared red and hair follicles across all groups were in the resting phase, with no significant morphological differences. By Day 7, no substantial changes in hair growth were observed in any group, although hair follicles had progressed into the anagen phase. The skin gradually darkened and took on a grayish hue, signaling the onset of anagen. Increased melanin production was apparent in all groups compared to Day 0, with no significant differences in the extent of skin darkening between groups.

By Day 14, more evident hair growth was observed in the minoxidil, Fr1, and Fr2 groups in contrast to the delayed growth in the model group. The Fr2 group exhibited more vigorous hair growth than the Fr1 group, which showed larger uncovered skin areas. By Day 21, hair growth had further increased in all groups. However, certain regions of the model group still had shorter hair with no visible growth, suggesting a shortened anagen phase in these mice compared to the experimental and positive control groups. The hair color of mice in both the control and Fr2 groups had returned to normal while the hair color in the Fr1 group remained darker, indicating a shorter growth period in the Fr1 group relative to the Fr2 group.

These observations demonstrate that intraperitoneal testosterone injections can delay hair growth and shorten the hair follicle growth period in mice. However, minoxidil and the prepared Fr1 and Fr2 fractions were able to counteract these adverse effects of testosterone, prolong the hair growth period, and promote hair growth.

#### 2.3.2. The Effect of Fr1 and Fr2 on Mouse Skin

##### Longitudinal Section Analysis of Mouse Skin

The results of the longitudinal section treatment of mouse skin and the changes in skin thickness (Figure 4A) indicated that the skin thickness on the backs of mice in the model group was significantly reduced at all treatment time points compared to the control group (*p* < 0.05). The skin in the model group appeared particularly atrophied. However, treatment with the compounds resulted in varying degrees of increased skin thickness, with the Fr2 group showing a particularly significant effect (*p* < 0.01). At all time points, the skin thickness in the Fr2 group was nearly equivalent to that of the normal group.

Hair follicles on the backs of mice in each group grew at varying rates over time. On Days 14 and 21, the growth rate of hair follicles in the model group was slower than that of the normal group. However, adding compounds with different compositions significantly enhanced the growth rate of hair follicles in the model group. Among these treatments, the Fr2 composition exhibited a remarkable promotive effect compared to the others.

##### Transverse Section Analysis of Mouse Skin

The results of the transverse section treatment of mouse skin and changes in hair follicle count (Figure 5A) showed that the number of hair follicles in all treatment groups was consistently around 50–55 on Day 0. Throughout the treatment period, the number of hair follicles in the model group was significantly reduced compared to the normal group (*p* < 0.01). However, the treatment groups promoted hair follicle growth to varying extents, with the Fr2 group exhibiting a significant increase in follicular growth and development. The Fr2 group reached a peak density of more than 80 hair follicles (*p* < 0.01).

Throughout the hair follicle growth cycle, the number of hair follicles increased at varying rates over time in all groups, except for the model group. In the Fr2 group, the number of hair follicles was close to that in the normal group, demonstrating a more pronounced effect in promoting hair follicle growth and development compared to the Fr1 group.

#### 2.3.3. Antioxidant Capacity Test

The impact of different treatment groups on antioxidant capacity was evaluated by measuring the activities of antioxidant enzymes (SOD, CAT, GSH-PX) and the content of the oxidative stress marker (MDA). The experimental results (Figure 6) revealed that antioxidant enzyme activities were significantly reduced in the model group. At the same time, MDA content, a marker of oxidative stress, was particularly increased, indicating a significant oxidative stress state in the model group.

SOD activity in the model group was significantly lower, decreasing from approximately 6 U/mg protein in the control group to around 4 U/mg protein. Similarly, CAT activity dropped considerably from about 3.5 U/mg protein in the control group to around 2 U/mg protein and GSH-PX activity decreased from approximately 4.5 U/mg protein in the control group to about 3 U/mg protein. However, the MDA content in the model group increased significantly from approximately 3 nmol/mg protein in the control group to about 4.5 nmol/mg protein (*p* < 0.05).

In comparison, the antioxidant enzyme activities in the Fr2 group were significantly higher than those in the model group, with SOD activity reaching approximately 5.5 U/mg protein, CAT activity around 3 U/mg protein, and GSH-PX activity about 5 U/mg protein in the Fr2 group. Furthermore, the MDA content in the Fr2 group was significantly lower than in the model group, at approximately 3 nmol/mg protein (*p* < 0.05). No significant differences in antioxidant enzyme activities were observed between the Fr1 and model groups.

In summary, the minoxidil and Fr2 groups significantly enhanced antioxidant enzyme activities and reduced MDA content, indicating that these two treatments had a marked ameliorating effect on oxidative stress. In comparison, the Fr1 group showed minimal impact on the experimental parameters and did not exhibit a significant antioxidant effect.

### 2.4. The Effects of DHHB on DPCs

#### 2.4.1. The Effects of DHHB on DPC Viability

The MTT assays (Figure 7B) demonstrated that DHHB had no significant effect on cell viability at concentrations below 5 µM. Cell viability was around 100% at 0 µM DHHB and approximately 95% and 90% at 2.5 µM and 5 µM DHHB, respectively. However, cell viability significantly decreased at DHHB concentrations above 10 µM. Based on these results, 5 µM DHHB was selected as the treatment concentration for subsequent experiments.

Cell morphology analysis (Figure 7A) showed that cells in the model group exhibited wrinkling and membrane rupture. Treatment with minoxidil restored cellular morphology, resulting in round cells of normal length. While 2.5 µM DHHB reduced cell death to some extent, its effects on cell morphology were limited. However, 5 µM DHHB significantly reduced DHT-induced cell death, improved morphological integrity, prevented membrane rupture, and promoted cell elongation.

Lactate dehydrogenase (LDH), a stable cytoplasmic enzyme, is released into the extracellular space only when the cell membrane is damaged. Therefore, extracellular LDH activity serves as an indirect indicator of membrane integrity [21]. As shown in Figure 7C, extracellular LDH activity in the model group was significantly higher than in the control group (*p* < 0.05). The minoxidil and DHHB treatment groups showed considerably reduced extracellular LDH activity compared to the model group (*p* < 0.05).

#### 2.4.2. The Expression of Related Proteins in DPCs

DHT binds to the androgen receptor (AR) with greater affinity than the biologically active form of testosterone. In dermal papilla cells (DPCs) (Figure 8A), DHT treatment led to a significant increase in AR protein expression compared to the control group (*p* < 0.05). However, minoxidil and 5 µM DHHB significantly suppressed AR protein levels relative to the model group (*p* < 0.05). Furthermore, intracellular levels of Transforming Growth Factor-β (TGF-β) were markedly elevated in the model group compared to the controls (*p* < 0.05) while treatment with minoxidil or DHHB significantly reduced TGF-β expression (*p* < 0.05) (Figure 8B).

The Wnt/β-catenin signaling pathway is crucial in various physiological and pathological processes. Extracellular Wnt ligands are highly conserved secreted proteins that bind to cell surface receptors and activate the canonical Wnt pathway via autocrine or paracrine signaling. Upon activation, Wnt signaling inhibits β-catenin degradation, allowing β-catenin to accumulate in the cytoplasm and translocate into the nucleus, regulating the expression of target genes involved in cell survival, proliferation, differentiation, and migration [22].

In the model group, Wnt protein levels were significantly reduced, while β-catenin levels were markedly elevated compared to the control group (*p* < 0.05), suggesting dysregulation of the pathway (Figure 8C,D). Treatment with minoxidil and 5 µM DHHB significantly restored Wnt levels and reduced β-catenin expression relative to the model group (*p* < 0.05). However, 2.5 µM DHHB only significantly increased Wnt expression without affecting β-catenin levels.

#### 2.4.3. The Effect of DHHB on mRNA Expression

Quantitative PCR (qPCR) is a sensitive and efficient technique for measuring mRNA expression levels. It involves the reverse transcription of unstable mRNA into complementary DNA (cDNA), followed by amplification and real-time fluorescence-based detection [23]. Relative gene expression is quantified by comparing the cycle threshold (Ct) values during amplification.

As shown in Figure 9, qPCR analysis revealed that AR mRNA expression in the model group was significantly elevated compared to the control group, with a relative expression level of approximately 7 (*p* < 0.05). Treatment with minoxidil and DHHB markedly suppressed AR mRNA expression. Relative expression levels were approximately 1.5 in the minoxidil group, about 3 in the 2.5 µM DHHB group, and around 2 in the 5 µM DHHB group (*p* < 0.05 vs. model group). These findings suggest that both minoxidil and DHHB effectively downregulate AR gene expression.

The relative expression of TGF-β mRNA was significantly elevated in the model group compared to the control group, reaching a value of approximately 3 (*p* < 0.05). TGF-β mRNA levels were reduced considerably in the minoxidil- and DHHB-treated groups. Relative expression levels were around 1.2 in the minoxidil group, approximately 2.5 in the 2.5 µM DHHB group, and about 1.5 in the 5 µM DHHB group (*p* < 0.05 vs. model group). These results indicate that both minoxidil and DHHB effectively suppressed TGF-β gene expression.

## 3. Discussion

*Platycladus orientalis* L. has traditionally been valued in Chinese medicine for its therapeutic properties, particularly its reputed ability to prevent hair loss and stimulate growth. A thorough review of the current scientific literature confirms that the biochemical effects of extracts such as volatile oils derived from *P. orientalis* [24]. However, the precise mechanisms driving these effects remain largely unclear and warrant further investigation. In this study, we specifically focused on the non-polar constituents of *P. orientalis* to evaluate their potential role in hair growth promotion and to clarify their underlying mechanisms of action.

Initially, low-polarity compounds were extracted from the leaves and branches of *P. orientalis* using n-hexane. The resulting extract was fractionated via medium-pressure liquid chromatography and the most bioactive fraction was identified through in vivo mouse assays. Subsequent purification using two-dimensional high-performance liquid chromatography led to the isolation of DHHB, a compound previously reported in *Biota orientalis* and *Heliopsis helianthoides* var. *scabra*. While DHHB has been noted for its neuroprotective effects in vitro, its potential biological applications, including hair growth promotion, have not been previously investigated.

Building on the observation that Fr2 significantly promotes hair growth in C57BL/6 mice, we further evaluated the bioactivity of DHHB using an in vitro model of AGA established by DHT-induced DPCs. Cells interact with their external environment through intricate signaling networks that regulate intracellular protein expression via modular functional domains [25]. The growth and proliferation of DPCs are modulated by various intracellular and extracellular factors, including androgen receptor activity, cell cycle dynamics, and key signaling pathways [26,27].

In the cytoplasm, testosterone is converted into DHT by the enzyme 5α-reductase. DHT then binds to the AR, forming a complex that translocates to the nucleus to modulate gene transcription and subsequent protein expression [28]. Cellular signaling is partly governed by secreted proteins, such as cytokines, growth factors, and hormones. TGF-β is a pivotal cytokine in maintaining tissue homeostasis and regulating developmental processes. Aberrant TGF-β signaling has been linked to various human pathologies [29]. In the context of AGA, research has demonstrated that androgens can directly activate the TGF-β promoter, highlighting TGF-β as a potential therapeutic target in AGA treatment strategies [30].

Comprehensive analyses using Western Blotting and qPCR revealed the multi-dimensional mechanism underlying DHHB’s activity. DHHB markedly suppressed the expression of the AR and TGF-β. DHHB was found to activate the Wnt/β-catenin signaling pathway, a key component of its mechanistic action. This finding is consistent with previous work by G.J. Leirós et al., who demonstrated that DPCs promote the differentiation of hair follicle stem cells via cytokine release in a co-culture model derived from AGA patients [31]. In that model, DHT inhibited Wnt/β-catenin signaling, therefore impairing the DPC-mediated induction of stem cell differentiation. These results indicate that DHHB facilitates DPC proliferation and mitigates DHT-induced follicular miniaturization by reactivating the Wnt/β-catenin pathway.

In conclusion, DHHB is a key bioactive constituent of *Platycladus orientalis* L., with promising potential as a novel agent for promoting hair growth. Further investigations are warranted to validate its therapeutic efficacy and elucidate its molecular mechanisms.

## 4. Materials and Methods

### 4.1. Instrumentation and Reagents

Preparative liquid chromatography was conducted using equipment from Hanbon Science & Technology Co. (Huai’an City, Jiangsu Province, China), which included two NP7000 prep-HPLC pumps, an NU3000 ultraviolet-visible (UV-Vis) detector, and a liquid chromatography (LC) workstation. Each system featured dual binary gradient pumps, an UV-Vis detector, a column oven, and an integrated workstation. The two HPLC units were connected via a triple valve and a polyether ether ketone (PEEK) reaction coil (18.0 m × 0.25 mm i.d.). Analytical HPLC was performed using the LC-16 system. Electrospray ionization mass spectrometry (ESI-MS) was carried out using a Waters QDa Mass Spectrometer (Waters Instruments Co., Milford, MA, USA). Proton (^1^H) and carbon (^13^C) nuclear magnetic resonance (NMR) spectra were recorded on a 600 MHz Bruker Avance spectrometer (Bruker Instruments Co., Karlsruhe, Germany), using DMSO-d_6_ as the solvent. UV absorbance was measured with a Readmax 1900 microplate reader (Flash Co., Shanghai, China).

Silica gel (100–200 mesh) was purchased from Qingdao Ocean Chemical Company (Qingdao, China). MCI GEL^®^ CHP20P (120 μm) was obtained from Mitsubishi Chemical Corporation (Tokyo, Japan) for chromatographic separation. A Diol column (50 × 500 mm, 25 μm) was sourced from ACCHROM Corporation (Beijing, China) and a spherical C18 column (50 × 500 mm, 50 μm) was acquired from SiliCycle (Québec City, QC, Canada). Furthermore, two ReproSil-Pur C18 AQ columns (4.6 × 250 mm, 5 μm, and 20 × 250 mm, 5 μm) were supplied by Dr. Maisch GmbH (Ammerbuch-Entlingen, Baden-Württemberg, Germany).

### 4.2. Plant Sample Preparation and Medium-Pressure Liquid Chromatography Pretreatment

*Platycladus orientalis* L. samples were collected in March 2021 from Yantai, Shandong Province, China (elevation: 806 m, coordinates: 37°28′ N, 120°83′ E) and were deposited in the School of Life Sciences, Yantai University.

A total of 550 g of *Platycladus orientalis* L. branches were air-dried in a well-ventilated area then ground into a fine powder. The powdered material underwent triple percolation extraction with hexane (5.5 L per extraction) for 10 h at room temperature. The combined percolates were concentrated under reduced pressure (100 mbar) at 40 °C using a rotary evaporator, resulting in approximately 60 mL of crude extract.

For medium-pressure liquid chromatography (MPLC) pretreatment, MCI GEL CHP20P (MCI), a medium-pressure chromatographic gel for separating natural organic small molecules, was used. A 30 mL aliquot of the crude extract was applied to a MPLC column (49 × 460 mm) packed with microporous MCI resin (stationary phase: CHP20P). Elution was carried out using chromatographically pure water (Mobile Phase A) and methanol (Mobile Phase B) at a flow rate of 50 mL/min, following the gradient: 0–240 min, 65–100% B; 240–300 min, 100% B. The total injection volume was 10 mL. The first and second major chromatographic peak fractions (Fr1 and Fr2) were collected with an UV detector set at 210 nm, followed by vacuum drying at 40 °C (100 mbar).

After MCI medium-pressure chromatographic pretreatment, the two components and the total samples of lateral cypress were prepared into 10 mg/mL sample solutions with methanol and filtered through a 0.45 μm organic membrane into liquid-phase vials for analysis. The HPLC analysis conditions were as follows: a Kromasil C18 AQ column (4.6 × 250 mm, 5 μm) was used, with Mobile Phase A being 0.1% formic acid aqueous solution and Mobile Phase B being chromatographic methanol. A gradient elution program was employed: 10%~100% B over 0~45 min, followed by 100% B from 45~60 min. The flow rate was set at 1 mL/min, the injection volume was 10 μL, and the detection was monitored using an UV detector set at 210 nm.

### 4.3. Liquid Chromatography Separation and Purification of Fr211 from Fr21

The dried Fr2 was dissolved in methanol to prepare a sample solution with a concentration ranging from 0.5 to 1.0 g/mL. This solution was filtered through a 0.45 μm microporous membrane to obtain the Fr2 filtrate. The filtrate was subjected to reversed-phase C18 liquid chromatography, using a water-resistant Kromasil C18 AQ column (250 × 20 mm, 5 μm) as the stationary phase. The mobile phase consisted of chromatographically pure water (A) and methanol (B) and elution was carried out at a flow rate of 19 mL/min following the gradient: 0–60 min, 65–85% B. The injection volume was 0.5 mL. The eluates were monitored using an UV detector set at 210 nm and the Fr21 chromatographic peak fractions were collected. These fractions were then vacuum-dried at 50 °C under reduced pressure (250 mbar) to obtain the target fraction.

The obtained Fr21 was dissolved in methanol at a concentration of 100.0 mg/mL and filtered through a 0.45 μm microporous membrane to obtain the Fr21 filtrate. This filtrate underwent further purification via reversed-phase Phenyl column liquid chromatography (RP-CPLC) using a Kromasil 100-5 Phenylphenyl column (250 × 20 mm, 5 μm) as the stationary phase. The mobile phase was composed of 56% methanol in water and the separation was carried out at a flow rate of 19 mL/min with an injection volume of 0.7 mL. The Fr211 fraction, corresponding to the peak in the reversed-phase Phenylphenyl column chromatogram, was collected using an UV detector set at 210 nm. The collected fraction was then vacuum-dried at 50 °C under reduced pressure (250 mbar), yielding 0.36 g of Fr211 with a purity exceeding 98%.

### 4.4. Establishment of AGA Model Mice

Specific pathogen-free (SPF) C57BL/6 male mice weighing 18 and 22 g were provided by Jinan Peng Yue Laboratory Animal Breeding Co. (Shandong, China). The mice were housed in a controlled environment with a relative humidity of 65–70%, a temperature range of 21–23 °C, and a 12-h light/dark cycle. They had ad libitum access to food and water throughout the experimental period. After one week of acclimatization, mice with similar growth characteristics were selected, weighed, and randomly assigned to groups based on body weight.

For depilation, mice were anesthetized with an intraperitoneal injection of 5% trichloroacetaldehyde hydrate. Heated wax was then cooled to a safe temperature and applied uniformly to a 3 cm × 4 cm area on the backs of the mice. Once the wax solidified, it was removed, completing the depilation process. To induce the androgenetic alopecia (AGA) model, testosterone propionate dissolved in olive oil was administered via intraperitoneal injection at a dose of 10 mg/kg once daily for 21 days.

To prepare the chloral hydrate solution, 5 g of chloral hydrate crystals were dissolved in 95 mL of distilled water under magnetic stirring at 20–25 °C in a fume hood. The solution was stored in a dark bottle at 4 °C for over one week. Furthermore, 20 mg of testosterone propionate was dissolved in 10 mL of sesame oil (or 10% ethanol/saline) to achieve a 2 mg/mL concentration. After vortexing until the solution was clear, it was filtered through a 0.22 μm syringe filter and administered at 0.1 mL per 10 g of body weight.

### 4.5. AGA Mice Experiment and Detection

Mice were randomly assigned to five groups (n = 10 per group) based on body weight. All groups, except the control group, received daily intraperitoneal injections of testosterone propionate as previously described. The control and model groups were treated with daily topical applications of olive oil, while the experimental groups were treated with 10% (*v*/*v*) solutions of Fr1 and Fr2. The positive control group received minoxidil (0.05 g/mL) treatment. All topical treatments were administered at a dose of 5 mL per kg of body weight.

Every 7 days, three mice from each group were randomly chosen for anesthesia and photographed to assess hair growth. On Days 7, 14, and 21 of the experiment, one mouse from each group was euthanized and longitudinal skin tissue strips (0.5 cm × 0.5 cm) were harvested from the same dorsal area. The tissues were either fixed in 4% paraformaldehyde for histopathological analysis or flash-frozen in liquid nitrogen and stored at −80 °C for subsequent biochemical assays. On day 21, all remaining mice were euthanized following the same procedure. Paraffin-embedded skin sections from the depilated dorsal region were prepared and stained with hematoxylin and eosin (H&E). The sections were examined under a 20× microscope to evaluate hair follicle count and growth status.

Dorsal depilated skin tissue samples from the experimental mouse groups, stored at −80 °C, were homogenized as follows: Approximately 50 mg of tissue was weighed, minced, and then homogenized in a 9× (volume/weight) pre-chilled PBS buffer using a cryogenic tissue homogenizer. The homogenate was centrifuged at 12,000 rpm for 15 min at 4 °C and the resulting supernatant was aliquoted for further analysis. Before biochemical assessments, the total protein concentrations of all samples were determined using the BCA assay (Thermo Fisher Scientific, Waltham, MA, USA). Oxidative stress markers, including superoxide dismutase (SOD) activity, malondialdehyde (MDA) content, catalase (CAT) activity, and glutathione peroxidase (GSH-Px) activity, were analyzed using commercially available assay kits (Nanjing Jiancheng Bioengineering Institute, Nanjing, Jiangsu Province, China) according to the manufacturers’ protocols. All procedures were meticulously standardized to ensure reproducibility and technical precision.

### 4.6. The Cell Viability of DHHB on DPCs

DPCs were obtained from iCell Bioscience Inc. (Shanghai, China) under catalog number iCell-0163a. These cells, derived from surgically excised normal scalp tissue, were confirmed positive for fibronectin through immunofluorescence staining. DPCs, which are fibroblast-like cells, grow in an adherent fashion. Low-passage DPCs, during the exponential growth phase, were seeded into 96-well plates at a density of 4 × 10^3^ cells per well and incubated for 24 h to facilitate cell attachment. Each experimental group included three replicate wells. Cells were treated with varying concentrations of DHHB for specified periods. Following treatment, 10 µL of MTT (3-(4,5-dimethylthiazol-2-yl)-2,5-diphenyltetrazolium bromide) solution was added to each well and the plates were incubated for a further 3 h. The culture medium was then carefully removed and 100 µL of dimethyl sulfoxide (DMSO) was added to each well to dissolve the formazan crystals. The plates were shaken for 15 min at room temperature to ensure complete dissolution. Absorbance (OD) was measured at 490 nm using a microplate reader. Each experiment was performed in triplicate. Cell viability was calculated using the following formula:$$Cell\;viability(\%)=\frac{{OD}_{experimental\;group}}{{OD}_{control\;group}}\times 100\%$$

### 4.7. The Morphological Changes of DHHB on DPCs

Following treatment with DHHB, morphological changes in DPCs were carefully examined under an inverted phase-contrast microscope at different magnifications. Representative images were captured to document these observations.

### 4.8. The Detection of LDH Activity

DPCs were plated in 12-well plates at a density of 4 × 10^4^ cells per well and allowed to adhere for 24 h. After attachment, the cells were treated with various concentrations of DHHB for 48 h. Following the treatment, the culture supernatants were collected and centrifuged at 1000× *g* for 5 min at 4 °C. The resulting supernatants were then used to measure lactate dehydrogenase (LDH) activity according to the manufacturer’s protocol using a commercial kit (Nanjing Jiancheng Bioengineering Institute). Each experiment was performed in triplicate.

### 4.9. The Detection of Protein Changes on DPCs

Total cellular proteins were extracted using cell lysis buffer (P0013, Beyotime, Shanghai, China) supplemented with 1% phenylmethylsulfonyl fluoride (PMSF; ST506-2, Beyotime, China). Protein concentrations were determined using the Enhanced BCA Protein Assay Kit (P0009, Beyotime, Shanghai, China). Equal amounts of protein were resolved by 10% SDS-PAGE and transferred to polyvinylidene difluoride (PVDF) membranes (ISEQ00010, Millipore, Tullagreen, Carrigtwohill, Ireland) at a constant current of 250 mA. The membranes were blocked with 5% non-fat milk (G5002, Servicebio, Wuhan, Hubei Province, China) for 1.5 h at room temperature and then washed three times with Tris-buffered saline containing Tween-20 (TBST). Following the blocking step, the membranes were incubated overnight at 4 °C with antibodies against β-actin, AR, TGF-β, Wnt, and β-catenin (Cell Signaling Technology, CST), with β-actin serving as a loading control. After three further washes with TBST, the membranes were incubated with a horseradish peroxidase (HRP)-conjugated secondary antibody for 1 h at room temperature. Protein bands were detected using a chemiluminescent substrate. Data are presented as the mean ± standard deviation (SD) of three independent experiments.

### 4.10. RT-PCR

DPCs in the logarithmic growth phase were collected, resuspended, and plated into 6-well plates at a density of 8 × 10^4^ cells/well for 24 h of incubation. After cell attachment, they were treated with compound formulations for 48 h. RNA extraction was performed after treatment: cells were washed with PBS, digested with EDTA-free trypsin, and centrifuged (1000× *g* for 5 min at 4 °C). The cell pellets were lysed with 300 μL lysis buffer, homogenized, and processed through a DNA removal column (13,400× *g* for 60 s at 4 °C). RNA was purified using ethanol precipitation, proteinase treatment, and multiple wash steps (13,400× *g* for 30–60 s at 4 °C) then eluted in 30 μL RNase-free water. RNA concentration was determined using UV spectrophotometry and normalized to 1 μg/μL. DNA contamination was removed using DNase I treatment (42 °C for 5 min), followed by cDNA synthesis (SPARKscript II RT Plus, 50 °C for 15 min, 85 °C for 5 min).

Specific primers for target genes were designed using Primer 5 software and synthesized by Shenggong Bioengineering (Shanghai, China) Co., Ltd. The primer sequences for the target proteins are listed in Table 1.

For RT-qPCR, primers (10 μM stock) and reaction mixtures were prepared using the SYBR Green master mix, 0.4 μL of primers, 1 μL of cDNA, and 7.8 μL of RNase-free water. Triplicate reactions were loaded into 8-tube strips, centrifuged, and subjected to standardized conditions: pre-denaturation at 95 °C for 600 s, followed by 40 cycles of denaturation at 95 °C for 15 s, annealing at 60 °C for 60 s, extension at 72 °C for 30 s, and melt curve analysis with a ramp rate of 0.5 °C/s from 70 to 95 °C. Data were analyzed using the 2^−ΔΔCt^ method, with ROX as the reference. Data are presented as the mean ± standard deviation (SD) of three independent experiments.

### 4.11. Statistical Analysis

All experiments were performed in triplicate and the results are presented as the mean ± standard deviation (SD). Statistical analyses were conducted using one-way analysis of variance (ANOVA) followed by Tukey’s test for multiple comparisons with SPSS version 20.0 software (SPSS Inc., Chicago, IL, USA). A *p*-value of <0.05 was considered statistically significant.

## 5. Conclusions

In conclusion, this study used medium-pressure liquid chromatography and two-dimensional high-pressure liquid chromatography to isolate active fractions from *Platycladus orientalis* L., focusing on evaluating their hair-growth-promoting activity in C57BL/6 mice. Subsequent separation and purification led to the identification of DHHB as a key active compound. In an in vitro DPC model, DHHB was demonstrated for the first time to counteract DHT-induced AGA. Mechanistic investigations, including Western Blotting and qPCR analysis, indicated that DHHB’s effects are closely linked to the Wnt/β-catenin signaling pathway.

These results provide theoretical and experimental support for further exploring *Platycladus orientalis* L. as a promising source of hair-growth-promoting agents. Furthermore, this study offers valuable insights into discovering novel compounds with potential hair-growth-enhancing properties.

## Figures and Tables

**Figure 1 ijms-26-05189-f001:**
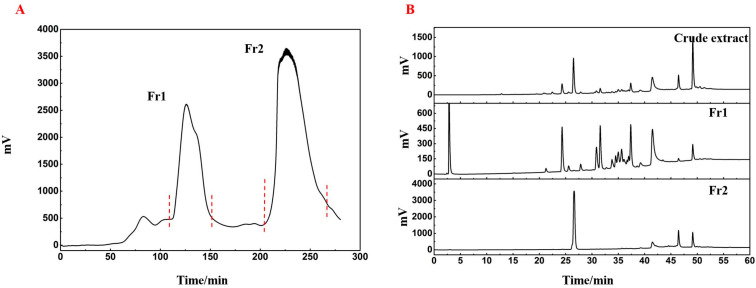
Sample pretreatment of *Platycladus orientalis* L. (**A**) Silica gel medium-pressure chromatogram showing separation of the *Platycladus orientalis* L. extract. (**B**) Analytical chromatograms for the two major crude extract fractions, designated Fr1 and Fr2.

**Figure 2 ijms-26-05189-f002:**
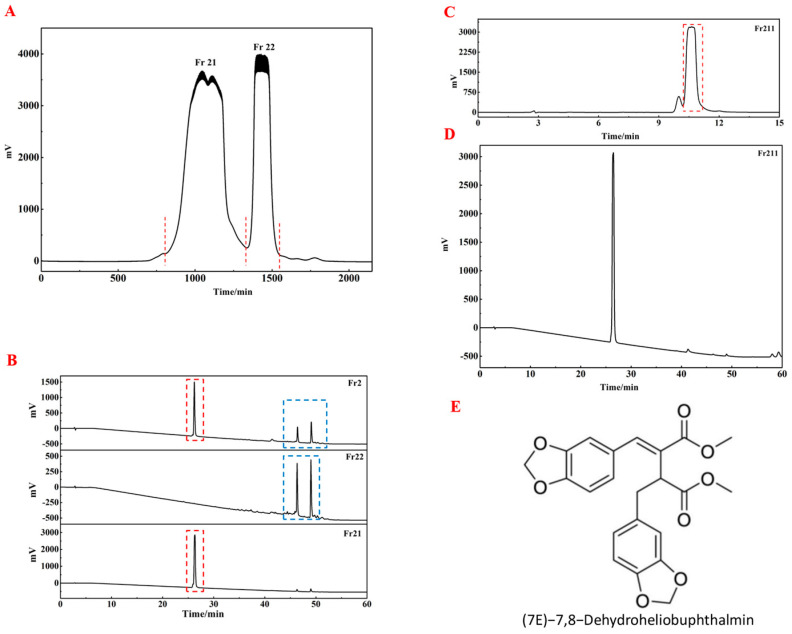
The isolation and purification of Fr211. (**A**) The pretreatment chromatogram for Fr2 using Kromasil C18 liquid chromatography. (**B**) Analytical chromatograms for Fr2, Fr21, and Fr22, respec tively. (**C**) The pretreatment chromatogram for Fr221 with Kromasil 100-5 Phenyl liquid chromatography. (**D**) Purity analysis of the isolated Fr211 on a Kromasil 100-5 Phenyl analytical column. (**E**) The chemical structure of Fr211 is identified as DHHB.

**Figure 3 ijms-26-05189-f003:**
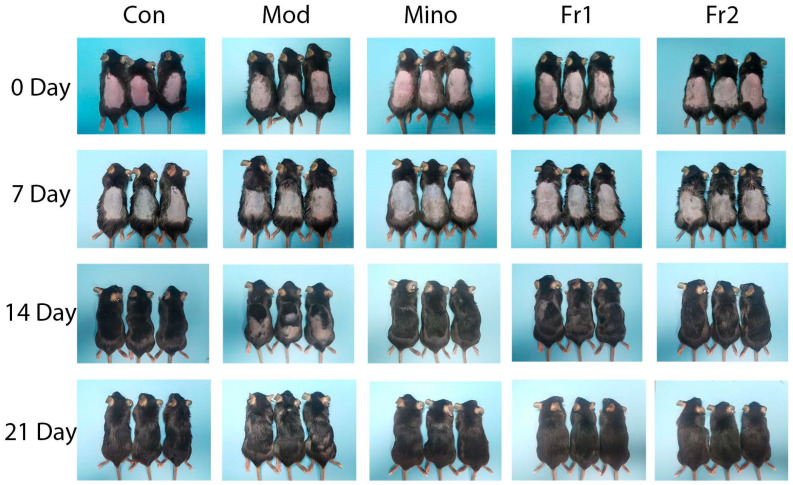
Hair growth on the back of C57BL/6 mice in various groups at different times.

**Figure 4 ijms-26-05189-f004:**
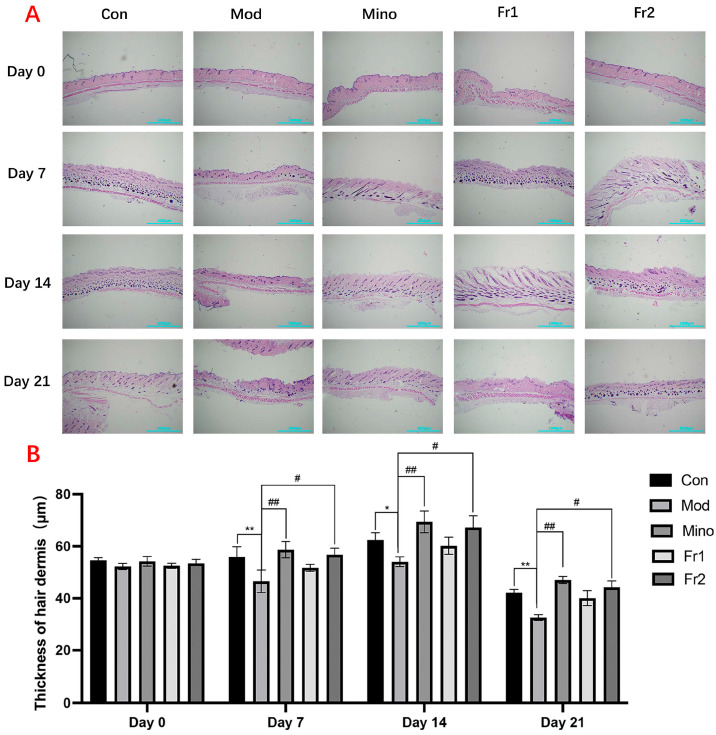
Longitudinal section analysis of mouse skin. (**A**) Representative longitudinal H&E-stained images showing back skin sections of different groups of C57BL/6 mice (*n* = 3). Scale bar = 1000 μm; (**B**) thickness of hair dermis in different groups of C57BL/6 mice (* *p* < 0.05; ** *p* < 0.01, model group vs. control group; # *p* < 0.05, ## *p* < 0.01, other groups vs. model group).

**Figure 5 ijms-26-05189-f005:**
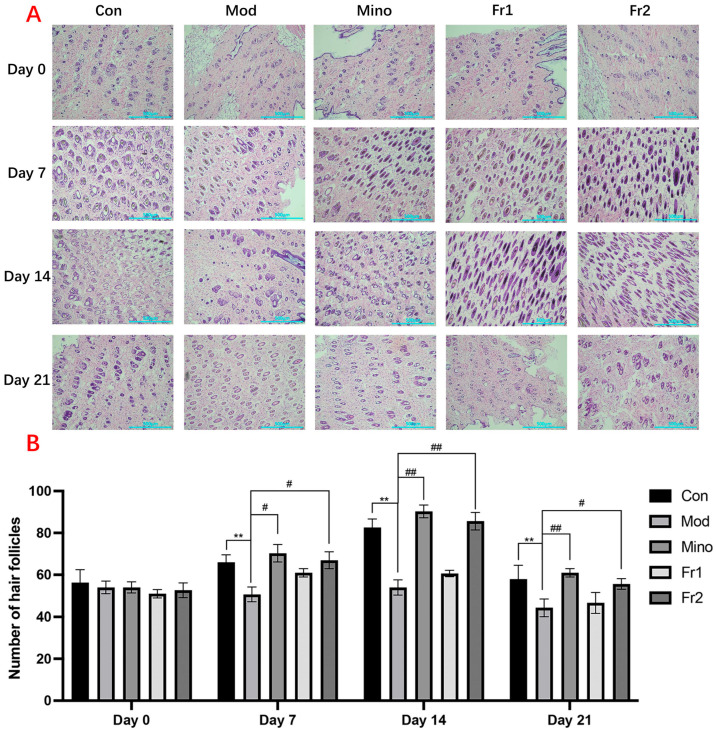
Transverse section analysis of mouse skin. (**A**) Representative transverse H&E staining images showing back skin sections from different groups of C57BL/6 mice (*n* = 3). Scale bar = 500 μm; (**B**) number of hair follicles in different groups of C57BL/6 mice (** *p* < 0.01, Model group vs. control group; # *p* < 0.05; ## *p* < 0.01, other groups vs. model group).

**Figure 6 ijms-26-05189-f006:**
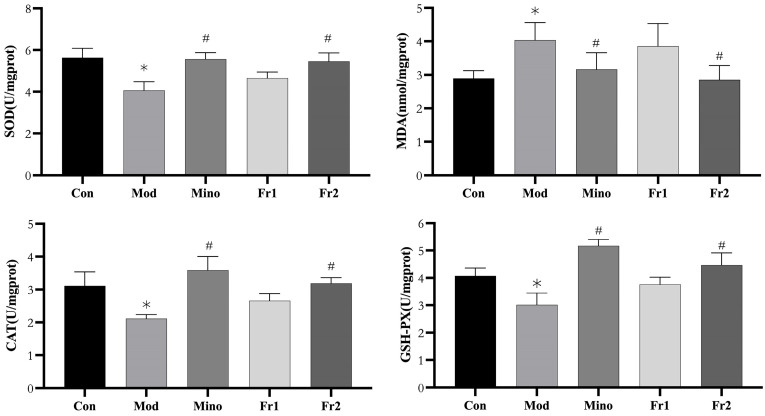
Antioxidant capacity of SOD, MDA, CAT, and GSH-PX in the back skin of C57BL/6 mice. (*n* = 3, * *p* < 0.05, model group vs. control group; # *p* < 0.05, other groups vs. model group).

**Figure 7 ijms-26-05189-f007:**
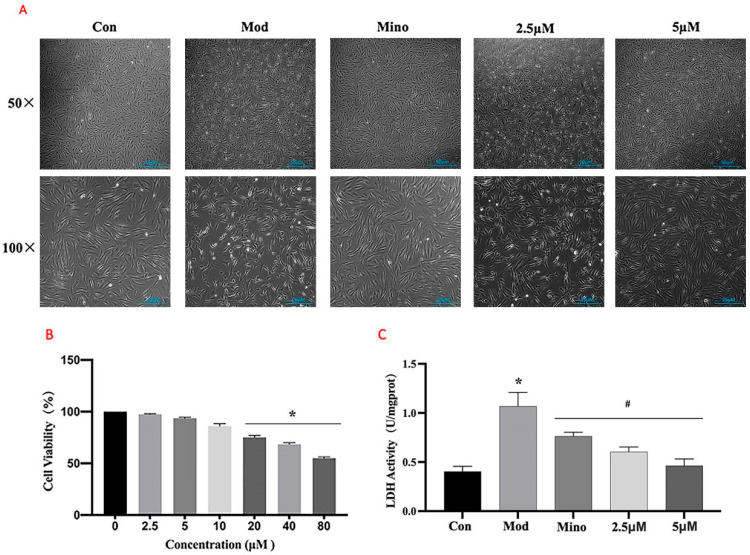
Effects of DHHB on DPC viability. (**A**) Effect of DHHB on cell morphology (Scale bar = 50 μm at 50× magnification, 25 μm at 100× magnification); (**B**) effect of DHHB on the cell viability; (**C**) LDH activity in the cytoplasm (* *p* < 0.05; model group vs. control group; # *p* < 0.05, other groups vs. model group).

**Figure 8 ijms-26-05189-f008:**
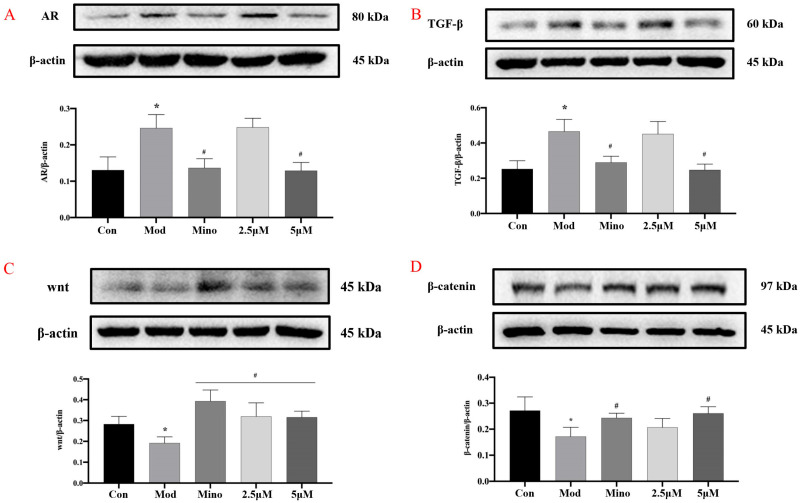
Effect of DHHB on cellular protein expression. (**A**) The expression of AR, (**B**), the expression of TGF-β, (**C**) the expression of protein Wnt, (**D**) the expression of β-catenin. (* *p* < 0.05; model group vs. control group; # *p* < 0.05, other groups vs. model group.).

**Figure 9 ijms-26-05189-f009:**
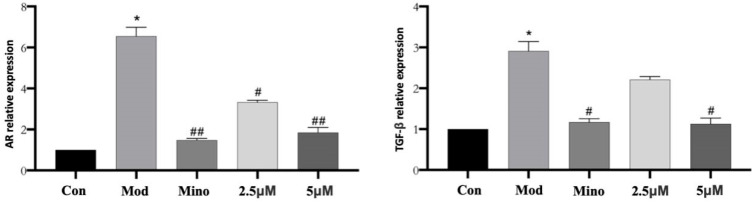
Effect of DHHB on the mRNA expression of ARs and TGF-β. (* *p* < 0.05, model group vs. control group; # *p* < 0.05, ## *p* < 0.01, other groups vs. model group).

**Table 1 ijms-26-05189-t001:** The primer sequence of related proteins.

Protein Name	(5′–3′)	(3′–5′)
GAPDH	TGAAGGTCGGAGTCAACGG	TGGAAGATGGTGATGGGAT
AR	GGGACCATGTTTTGCCCATT	GCAGCTTCCACATGTGAGAG
TGF-β	AGACTTTTCCCCAGACCTCG	TGGGTGGTCTTGAATAGGGG

## Data Availability

The data presented in this study are available in the article and the Appendix A.

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
