# Peer review of "(7E)-7,8-Dehydroheliobuphthalmin from *Platycladus orientalis* L.: Isolation, Characterization, and Hair Growth Promotion"

_ijms, 2025, doi:10.3390/ijms26115189_

Round 1
Reviewer 1 Report (New Reviewer)
Comments and Suggestions for Authors
This study focuses on the mechanism by which the medicinal properties Platycladus orientalis L. promote hair darkening and regeneration. These findings suggest that DHHB is a key active component of Platycladus orientalis L. and holds promise as a novel therapeutic agent for hair growth promotion.
The paper provides very interesting, but it still needs considerable revision to be acceptable. It is as follows.
(1) In this paper, it is written that the statistical analysis was performed using one-way ANOVA or Student's t-test to test significant differences, but I think that Student's t-test is not appropriate when examining three or more groups. I think it would be better to re-run the test using a more appropriate method.
(2) In 2.3.3 of the “Results”, it is unclear how the activity of antioxidant enzymes and the measurement of the oxidative stress marker were analyzed, but if it has been shown that the Fr2 group has antioxidant properties, I think it is necessary to discuss in the “Discussion” whether this effect is related to the inhibition of apoptosis.
(3) In this paper, the authors suggested that DHHB inhibited apoptosis, but the cell death shown in Figure 7 cannot be determined to be apoptosis. In the photographs of the cells in Figure 7 (A), it is not possible to determine whether the cell is apoptotic or not. I think that changes such as nuclear shrinkage and fragmentation characteristics of apoptosis should be confirmed, such as staining the nucleus with fluorescent dye. In addition, I don't think apoptosis can be detected by MTT assay or LDH assay. Also, I think it is difficult to prove apoptosis by the expression of Bcl-2 and BAX alone.
Minor points
(4) In 2.3.2 and 2.3.3 of the “Results”, explanations of the data were written, but it was not stated which figure they referred to. This should be clarified.
(5) In Figure 4, 5 and 7, the scale bars in the photos are unclear. Please include a clear scale bar in the photos.
(6) Please describe the number of experiments. The n number should be described in Figures 7,8 and 9.

Author Response
Point 1: In this paper, it is written that the statistical analysis was performed using one-way ANOVA or Student's t-test to test significant differences, but I think that Student's t-test is not appropriate when examining three or more groups. I think it would be better to re-run the test using a more appropriate method.
Response 1: Thank you for your suggestion. We agree ANOVA is more rigorous for such experimental designs. In the revised manuscript, all multi-group comparisons (e.g., Fig. 3-9) have been re-analyzed using one-way ANOVA with Tukey’s post-hoc test, and significance markers (*p<0.05, **p<0.01) have been updated accordingly. And, the Methods section now explicitly states: “Statistical analyses were conducted using either one-way analysis of variance (ANOVA) followed by Tukey’s test for multiple comparisons with SPSS version 20.0 software (SPSS Inc., Chicago, IL, USA).”
Point 2: In 2.3.3 of the “Results”, it is unclear how the activity of antioxidant enzymes and the measurement of the oxidative stress marker were analyzed, but if it has been shown that the Fr2 group has antioxidant properties, I think it is necessary to discuss in the “Discussion” whether this effect is related to the inhibition of apoptosis.
Response 2: We thank the reviewer for highlighting this important mechanistic point. After comprehensive consideration, we have decided to delete the research content related to apoptosis in the revised manuscript (e.g., Fig. 8-E). In the corresponding discussion section, we have also made certain modifications and adjustments. Apoptosis and the mechanism of DHHB will be the focus of our future research.
Point 3: In this paper, the authors suggested that DHHB inhibited apoptosis, but the cell death shown in Figure 7 cannot be determined to be apoptosis. In the photographs of the cells in Figure 7 (A), it is not possible to determine whether the cell is apoptotic or not. I think that changes such as nuclear shrinkage and fragmentation characteristics of apoptosis should be confirmed, such as staining the nucleus with fluorescent dye. In addition, I don't think apoptosis can be detected by MTT assay or LDH assay. Also, I think it is difficult to prove apoptosis by the expression of Bcl-2 and BAX alone.
Response 3: Thank you for your professional advice. Indeed, as you pointed out, the evidence for apoptosis in our manuscript is not very strong, and its relationship with the antioxidant capacity of DHHB has not been directly confirmed. After comprehensive consideration, we have decided to delete the research content related to apoptosis in the revised manuscript (e.g., Fig. 8-E) to better ensure the integrity of the paper. More relevant content about apoptosis will be carried out in the future to confirm the mechanism. Thank you again for your suggestion.
Point 4: In 2.3.2 and 2.3.3 of the “Results”, explanations of the data were written, but it was notstated which figure they referred to. This should be clarified.
Response 4: Thank you for your suggestion. We have compared the experimental results with the experimental images and have labeled lines 153 and 171 in the text, and the labeled sections have been highlighted in yellow.
Point 5: In Figure 4, 5 and 7, the scale bars in the photos are unclear. Please include a clear scale bar in the photos.
Response 5: Thank you for your professional advice. We have upgraded the quality of the image and pointed out the scale in the image caption section (lines 166, 183 and 233).
Point 6: Please describe the number of experiments. The n number should be described in Figures7,8 and 9.
Response 6: We are very grateful for your careful suggestion, and we have added experimental replicates in the Experimental Methods section, labeled in the article with a yellow background (lines 452, 465, 481, and 505), which have a very important impact on the completeness of the whole article.

Reviewer 2 Report (New Reviewer)
Comments and Suggestions for Authors
This study presents a novel approach to “(7E)-7,8-Dehydroheliobuphthalmin from Platycladus orientalis L.: Isolation, Characterization and Hair Growth Promotion.” The author has justified their study in multiple approaches with in-depth supportive literature. This study could be beneficial for many researchers. However, further discussion on a practical application basis could enhance the strength of your paper:
Q1. The initial line is abstract is not more supportive for a new study, making it stronger.
Q2. In abstract, highlight more about the results in numbers compared to the androgen model with your best samples.
Q3. In the introduction, the author mentions the role of DHT in hair loss. However, the author needs to highlight the novelty of their selected medicinal herbs compared to already established DHT-lowering agents such as finasteride.
Q4. Unify the Figure 4 skin image (Day 21 of control, Day 7 of FR2) as other group.
Q5. Its hard to differentiate between the in vitro and in vivo study. Please make in vitro study at first, followed by an in vivo and analysis data of in vivo (such as H&E staining).
Q6. If you define the abbreviation once in the beginning, use it all over the manuscript (eg. AGA: androgenic alopecia, HDP: dermal papilla cells).
Q7. Separately provide the abbreviation list after conflict of interest and before reference.
Q8. In method section 4.4 and 4.5, describe in detail about preparation of 5% trichloroacetaldehyde hydrate as well as the preparation of testosterone propionate and your sample solutions. Since the used solvents is important to determine the toxicity in mice.
Q9. Make clear scale bar in figure 7A.
Q10. In figure somewhere, its minoxidil, somewhere min, control or Con. Please unify it all. And make DHHB (2.5 µM) instead of 2.5 µM only. Also add dose of minoxidil, which will be easy to understand.
Author Response
Point 1: The initial line is abstract is not more supportive for a new study, making it stronger.
Response 1: Thank you for your suggestion. We have updated the first sentence of the abstract in lines 15-18 of the article to ensure the novelty of the literature.
Point 2: In abstract, highlight more about the results in numbers compared to the androgen model with your best samples.
Response 2: We are very grateful for your suggestion, and we have standardized lines 24-28 in the article to emphasize the comparison of the results compared to the model group, highlighting the significant effect of its action. This suggestion has helped us a lot by making the structure of the article clearer and more explicit.
Point 3: In the introduction, the author mentions the role of DHT in hair loss. However, the author needs to highlight the novelty of their selected medicinal herbs compared to already established DHT-lowering agents such as finasteride.
Response 3: Thanks to your suggestion. We have updated the introduction section to include in lines 67-69 the advantages that sidereal has over commercially available positive therapeutic drugs. This makes the article more logical and complete.
Point 4: Unify the Figure 4 skin image (Day 21 of control, Day 7 of FR2) as other group.
Response 4: Thank you for your suggestion, we have updated the Day 21 of control and Day 7 of FR2 images to replace them with the appropriate ones.
Point 5: Its hard to differentiate between the in vitro and in vivo study. Please make in vitro study at first, followed by an in vivo and analysis data of in vivo (such as H&E staining).
Response 5: Thanks a lot for your careful advices. Indeed, as you pointed out, the research work of this article is slightly confused in the first half of the content. First, we obtained two Fractions of Platycladus platycladus (Figure 1), then the germinative activity of the Fr2 was determined through a mouse AGA model (Figures 3-5), and the compound DHHB was obtained by multi-dimensional chromatography combination (Figure 2). Subsequently, cell experiments were conducted to study the germinative activity and preliminary mechanism of DHHB(Figures 6-8).
After comprehensive consideration, during the process of compiling the draft, we completed it in the order of chemical separation first, then in vivo and analysis, and finally in vitro study. Thank you again for your suggestion.
Point 6: If you define the abbreviation once in the beginning, use it all over the manuscript (eg. AGA: androgenic alopecia, HDP: dermal papilla cells).
Response 6: We appreciate your suggestions and we have carefully examined the full text and will rectify the abbreviations first suggested.
Point 7: Separately provide the abbreviation list after conflict of interest and before reference.
Response 7: Thank you for your suggestion and we have added a table of abbreviations between “Conflict of interest” and “References” (line 539) as requested.
Point 8:In method section 4.4 and 4.5, describe in detail about preparation of 5% trichloroacetaldehyde hydrate as well as the preparation of testosterone propionate and your sample solutions. Since the used solvents is important to determine the toxicity in mice.
Response 8: We are very grateful for your suggestion and we have added the detailed preparation of the corresponding solution in section 4.4 as requested and highlighted in yellow (lines 404-409). Once again, thank you very much for your suggestion, which is constructive for us.
Point 9:Make clear scale bar in figure 7A.
Response 9: Thank you for your suggestion, we have upgraded the image quality of figure 7A and pointed out the scale bar in the image explanation section.
Point 10: In figure somewhere, its minoxidil, somewhere min, control or Con. Please unify it all. And make DHHB (2.5 µM) instead of 2.5 µM only. Also add dose of minoxidil, which will be easy to understand.
Response 10: We appreciate your suggestion and we have stylized the resultant images and modified them as you requested. This is very important for the integrity of the article.

Round 2
Reviewer 1 Report (New Reviewer)
Comments and Suggestions for Authors
I think this paper has been well revised, but I think it needs some further revisions before it can be accepted.
Re-comment on Response 1:
I would like to confirm: In this study, the authors have changed the method of statistical analysis. I think that if the testing method changes, the test results will also change. The test results seem to be the same as last time, are they truly the same?
Re-comment on Response 2:
I have reviewed your response and understand that you will focus on apoptosis and the mechanism of DHHB in future research.
I would like to ask the authors to respond again: how was the antioxidant capacity of SOD, MDA, CAT and GSH-PX in the back skin of C57BL/6 mice measured? I would appreciate it if you could specify the name of the analysis method.
Re-comment on Response 5:
The scale bar in Figure 7A is still unclear. Please make it visible.

Author Response
Point 1: I would like to confirm: In this study, the authors have changed the method of statistical analysis. I think that if the testing method changes, the test results will also change. The test results seem to be the same as last time, are they truly the same?
Response 1:
Thank you for your professional advice. Indeed, as you pointed out, in the last revised manuscript, we only changed the description of statistical analysis in method section. This time, we re-analyzed our data using the new method and updated the corresponding graph in the main text (e.g., Fig.4, Fig.5, Fig.7, and Fig.9).
Point 2: I have reviewed your response and understand that you will focus on apoptosis and the mechanism of DHHB in future research.
I would like to ask the authors to respond again: how was the antioxidant capacity of SOD, MDA, CAT and GSH-PX in the back skin of C57BL/6 mice measured? I would appreciate it if you could specify the name of the analysis method.
Response 2:
Thank you very much for your approval, we will study the relationship between DHHB and the apoptosis in the future, which we think your advice will be constructive for the research.
Regarding the indicators you mentioned for detecting the antioxidant capacity of the dorsal skin, we gave a brief description in Section 4.5 of the main text (lines 426 -437). Since the reagent kits we used were commercially sold, we think it unnecessary to describe in detail the determination methods, principles and operational details of various indicators in this manuscript. We hope you can understand. The following are the methods and principles for each indicator.
The activity of superoxide dismutase (SOD) is determined using the hydroxylamine method. This method employs a xanthine/xanthine oxidase reaction system to generate superoxide anion radicals (O₂⁻), which subsequently oxidize hydroxylamine to form nitrite. The nitrite reacts with a chromogenic agent to produce a purple-red coloration, and the absorbance is measured using a visible spectrophotometer.
Malondialdehyde (MDA) content is assayed via the thiobarbituric acid (TBA) method. MDA, a degradation product of lipid peroxidation, condenses with TBA to form a red-colored adduct with a maximum absorption peak at 532 nm.
Catalase (CAT) activity is quantified through the ammonium molybdate method. The enzymatic decomposition of H₂O₂ by CAT is rapidly terminated by the addition of ammonium molybdate. The residual H₂O₂ reacts with ammonium molybdate to generate a pale-yellow complex, and CAT activity is calculated based on absorbance changes measured at 405 nm.
Glutathione peroxidase (GSH-Px) activity is determined by measuring its catalytic efficiency in promoting the reaction between hydrogen peroxide (H₂O₂) and reduced glutathione (GSH), yielding water (H₂O) and oxidized glutathione (GSSG). The enzymatic activity is quantified by assessing the rate of GSH consumption during the enzyme-catalyzed reaction.
We hope that this response fulfills your suggestions and thank you again for your help in this paper.
Point 3: The scale bar in Figure 7A is still unclear. Please make it visible.
Response 3:
Thank you very much. We appreciate your suggestion and have re-normalized Figure 7A to make the scale clear. This makes the figure more convincing and the article more complete.

This manuscript is a resubmission of an earlier submission. The following is a list of the peer review reports and author responses from that submission.
Round 1
Reviewer 1 Report
Comments and Suggestions for Authors
In the presented manuscript, the Authors decided to assess the effect of Platycladus orientalis L. extract on hair growth in two models, mouse and cell line. At the same time, they presented a method of isolating the component using two-dimensional high-pressure liquid chromatography.
The authors presented a clear hypothesis and designed the study very well, which which led to very promising conclusions.
As minor remarks, the Authors need to expand the discussion. Considering such extensive experimental work, the obtained results should be discussed in a much broader way. The Authors should also enrich the discussion with more up-to-date literature. Additionally, the materials and methods section should be expanded. There are missing necessary methodological details, such as information on the antibodies used or the PCR reaction profile.
Author Response
We sincerely appreciate your insightful and constructive suggestions, which have greatly improved our manuscript. In response to your comments, we have carefully revised the text as follows:
Discussion: We have thoroughly rewritten this section to provide a more comprehensive analysis, incorporating multiple perspective.
References: We have updated and expanded the reference list to ensure the rigor and completeness of our research.
Method: We have refined the general descriptions of the experimental procedures, ensuring greater precision. Where applicable, we have also added quantitative details to enhance clarity.
All modifications in the manuscript have been highlighted with a yellow background for your convenience.
Thank you again for your help and valuable feedback, which have significantly strengthened our work.

Reviewer 2 Report
Comments and Suggestions for Authors Lin et al attempted to study the effects of fraction2 of Platycladus orientalis in cells and mouse model of androgenetic alopecia. Major concerns: No data is presented to support activity of Dehydroheliobuphthalmin in the mouse model 80: provide equipment and method on how analytical chromatograms were generated 98: provide equipment and method on how NMR and ESI-MS data were generated\\ 143-144: the statement of superiority of F2 over F1 is not substantiated, either provide the quantitative data that support this stamtement or remove. Please also provide quantitative data that compare F2 activity to control (mod) and reference (mino) groups. 148-166: please remove the general description that has no meaning and describe the actual quantifiable data and statistical significance for each 7,14,21 day checkpoint. Currently this reads as AI-generated text. 173-191: please remove the general description that has no meaning and describe the actual quantifiable data and statistical significance for each 7,14,21 day checkpoint. Currently this reads as AI-generated text. 197-206: please remove the general description that has no meaning and describe the actual quantifiable data and statistical significance for all 4 markers. Currently this reads as AI-generated text. 212-226: please remove the general description that has no meaning and describe the actual quantifiable data and statistical significance for MTT. Currently this reads as AI-generated text. 232-250: most of information here is not results, this needs to be moved to introduction or discussion sections. Discuss quantifiable data. Currently this reads as AI-generated text. 265-273: please remove the general description that has no meaning and describe the actual quantifiable data and statistical significance. Currently this reads as AI-generated text. 290: include proper citations to previously isolated compound 291: include proper citations to neuroprotective activity 293: to support this statement, show quantifiable data that are significantly different for F2 300: Provide methods for western blot analysis and antibody sources in the methods section 293-309: include a discussion of the actual results and not a general meaningless statement. Currently this reads as AI-generated text. 403-406: provide the buffer for homogenization and the methods for SOD, MDA, CAT, and GSH activities 466-450: provide methods for RNA extraction, RT, and qPCR parameters Minor comments: Lines 354-356: delete the duplicated text 386: please indicate for how many days 408: provide the cell line name or the catalog number from iCell
Author Response
Dear reviewer,
Thank you very much for your suggestion to our manuscript (ijms-3520137). In response to your comments, we have carefully revised the text and all the modifications have been highlighted with a yellow background for your convenience.
We confirm that this manuscript is entirely original and has not been generated by artificial intelligence. To ensure the highest quality of academic English, the manuscript underwent professional language polishing by Lansen Editing Co., a specialized editing service, prior to submission. And we have uploaded the article manuscript and the attached supplementary materials.
In addition, we have uploaded relevant supplementary information and supporting materials at the end of this document.
Thank you again for your comments and help!
Sincerely
Lin Zikai

Round 2
Reviewer 1 Report
Comments and Suggestions for Authors
The Authors improved the manuscript.